# SERGHEI v2.1: a Lagrangian Model for Passive Particle Transport using a 2D Shallow Water Model (SERGHEI-LPT)

Pablo Vallés<sup>a,b</sup>, Mario Morales-Hernández<sup>a</sup>, Volker Roeber<sup>b</sup>, Pilar García-Navarro<sup>a</sup>, and Daniel Caviedes-Voullième<sup>c,c2</sup>

Correspondence: Daniel Caviedes-Voullième (d.caviedes.voullieme@fz-juelich.de)

Abstract. This paper presents a Lagrangian model for particle transport driven by a 2D shallow water model, assuming that the particles have negligible mass and volume, are located at the free surface, and without interactions between them. Particle motion is based on advection and turbulent diffusion, which is added using a random-walk model. The equations for particle advective transport are solved using the flow velocity provided by a 2D shallow water solver and an online first-order Euler method, an online fourth order Runge-Kutta method and an offline fourth order Runge-Kutta method. The primary objective of this work is to present the capabilities of the new Lagrangian particle model, while also providing an analysis of the accuracy and computational efficiency of the numerical schemes and their implementation for particle transport. To verify the accuracy and computational cost, several test cases inspired by laboratory setups are simulated. In this analysis, the Euler online method provides the best compromise between accuracy and computational efficiency. Finally, a localized precipitation event in the Arnás catchment is simulated to test the model's capability to represent particle transport in overland flow over irregular topography.

#### 1 Introduction

Over the past decade, floods have emerged as the most destructive natural disaster worldwide (Wallemacq et al., 2015; for Research on the Epidemiology of Disasters, CRED), resulting in substantial and progressively increasing economic losses (Formetta and Feyen, 2019; Ripple et al., 2020). Projections suggest that in the coming decades, both economic losses and the number of individuals affected by floods are expected to rise (Dottori et al., 2018; OECD, 2016). This situation forces institutions to develop strategies and tools to predict and mitigate the damage caused by floods. Potential solutions range from administrative policies based on practical actions (Nations, 2015; Olcina et al., 2016; Martin et al., 2010) to the development of computational prediction tools (Bates et al., 2023; Thielen et al., 2009; Chen et al., 2009). In recent years, these predictive tools, which rely on numerical simulations, have seen significant advancements in both accuracy and computational efficiency (Sampson et al., 2015; Knijff et al., 2010; Lacasta et al., 2014). Some models are based on the two-dimensional Shallow Water

<sup>&</sup>lt;sup>a</sup>I3A, University of Zaragoza, Zaragoza (Spain)

<sup>&</sup>lt;sup>b</sup>E2S Chair HPC-Waves, University of Pau, Anglet (France)

<sup>&</sup>lt;sup>c</sup>Institute of Bio- and Geosciences: Agrosphere (IBG-3), Forschungszentrum Jülich, Jülich (Germany)

<sup>&</sup>lt;sup>c2</sup>Simulation and Data Lab. Terrestrial Systems, Jülich Supercomputing Centre (JSC), Forschungszentrum Jülich, Jülich (Germany)

Equations (SWE), offering accurate results with high computational efficiency (Xia et al., 2019; Morales-Hernández et al., 2021). However, the majority of models used for flood forecasting in rivers do not account for the transport of objects, such as vehicles, waste containers and other urban flood drifters (Bayón et al., 2024), or wood, carried by the flow during flooding events. This omission reduces the completeness of these tools, as object transport is a critical component of floods and can cause obstructions in hydraulic structures within channels and streams, thereby exacerbating the damage during such events (Bayón et al., 2024; Valero et al., 2024; Lofty et al., 2024). Several studies have examined debris accumulation near weirs, culverts, and dams, highlighting their significant impact on flood risk (Thomas and Nisbet, 2012; De Cicco et al., 2015). Moreover, the transport of objects by geophysical flows is related to other issues, such as environmental accidents due to pollutant spills in rivers and lakes (van Emmerik et al., 2022; Ivshina et al., 2015; Mellink et al., 2024), and environmental impact studies concerning seed dispersion by precipitation in catchments (Zamora and Montagnini, 2007; Sánchez-Salas et al., 2017), among others.

The transport of objects by fluid flows has been extensively studied in recent years (Braudrick and Grant, 2000; Xia et al., 2011; Martínez-Gomariz et al., 2020). These studies have led to the development of computational models for simulating the transport of particles with negligible mass or volume (Finaud-Guyot et al., 2023), as well as contaminants and small objects such as microplastics (García-Martínez and Flores-Tovar, 1999; Jalón-Rojas et al., 2019), sediments, bedload transport (Zhao et al., 2024; Baharvand et al., 2023), and macroscopic objects (Persi et al., 2018a, b). Debris transport can be analyzed using either an Eulerian or a Lagrangian framework (Nordam et al., 2023). The Lagrangian approach primarily provides information on the pathways linking the origin to the destination of individual particles. It may also capture specific processes affecting debris, such as deposition, fragmentation, and degradation, provided that these processes are explicitly implemented. On the other hand, the Eulerian approach provides a different approach by treating debris as a concentration, offering greater computational efficiency than the Lagrangian method when the number of bodies is really high. However, the Eulerian description is inherently dispersive and does not account for the specific transport trajectories of particles, thereby overlooking certain small-scale details (Cai et al., 2023).

In recent decades, numerous computational models have been developed to simulate particle transport. However, the majority of these models are designed for ocean and sometimes coastal scenarios with fixed wet/dry boundaries (Lebreton et al., 2012; Liubartseva et al., 2018). Furthermore, some models update particle positions only at specific time intervals rather than at every time step, in order to reduce the high computational cost (Finaud-Guyot et al., 2023). These so-called offline methods are often used in coastal environments where the flow evolves on time scales much longer than the particle time step, making temporal interpolation feasible (Cucco et al., 2009; Fajardo-Urbina et al., 2023, 2024). It is important to note that flooding and drying occur in many coastal systems, such as estuaries or tidal flats, and offline methods have still been successfully applied in such contexts. The inaccuracy introduced by not updating particle positions at every step is often mitigated by using higher-order schemes, such as a fourth-order Runge-Kutta method (García-Martínez and Flores-Tovar, 1999). An alternative approach is to determine the temporal evolution of material by solving the Eulerian advection-dispersion equation, thereby obtaining the changes in debris concentration over time (Baharvand et al., 2023; Schreyers et al., 2024; Portillo De Arbeloa and Marzadri, 2024). While these models effectively capture the behavior of large quantities of small debris, they are not suitable for large objects or scenarios involving a small number of material particles. Additionally, some Lagrangian models are implemented

using three-dimensional hydrodynamic frameworks (Jalón-Rojas et al., 2019; Pilechi et al., 2022), which provide more detailed flow information compared to two-dimensional and one-dimensional models, albeit at the cost of increased computational time.

In addition to advection, turbulent diffusion can play an important role in the material transport, not only in water (Merritt and Wohl, 2002; Molazadeh et al., 2024; Yang and Foroutan, 2023), but also in other environments such as wind (Horn et al., 2012). Thus, it is interesting to add to the Lagrangian model of material transport some kind of minimal turbulence model to achieve greater realism and accuracy in the numerical results (Jalón-Rojas et al., 2019; Liubartseva et al., 2018).

Finally, in order to be effective, the computational model must be both accurate and computationally efficient. This balance is especially important in applications such as flood forecasting, real-time decision support, environmental impact assessments, and large-scale scenario simulations, where timely and reliable results are essential. In this context, depth-averaged two-dimensional hydrodynamic models have proven to offer a favorable trade-off, providing sufficient accuracy for many surface water flow scenarios while incurring significantly lower computational costs compared to fully three-dimensional models (Vacondio et al., 2016; Echeverribar et al., 2019).

This work represents an initial step towards a Lagrangian model for material transport in shallow water flows. The material simulated in this study are particles with negligible mass and volume that do not interact with each other, thus representing a simplified case. The primary objective is to present the capabilities of the new Lagrangian particle model, together with an accompanying analysis of the accuracy and computational efficiency of the numerical scheme and its implementation for particle transport. The accuracy and efficiency of an explicit Euler method are compared with those of a fourth order Runge-Kutta method. Additionally, these methods are evaluated both when applied at every time step or iteration, and at specific iteration intervals. Furthermore, particles dispersion is incorporated into the Lagrangian Particle Transport (LPT) model equations using a random-walk model because the hydrodynamic model does not resolve turbulence. All analyzes are based on simulations of both analytical and laboratory test cases and realistic scenarios. The model is driven by an Eulerian hydrodynamic framework based on the SWE to describe flow evolution. The implementation of these techniques is part of an ongoing initiative to create a modeling framework grounded in physically-based hydrodynamics: the SERGHEI (Simulation Environment for Geomorphology, Hydrodynamics, and Ecohydrology in Integrated form) model (Caviedes-Voullième et al., 2023). The code developed for this study is open-source and available at https://gitlab.com/serghei-model.

The structure of the paper is as follows: first, the governing equations for the hydrodynamic model and for the LPT model are presented; second, the numerical schemes and their details are presented; then, analytical, test and realistic cases are simulated; and finally, the conclusions of the present work are shown.

## 85 2 Governing equations

The flow equations are presented and subsequently, the equations corresponding to passive particle transport are discussed.

#### 2.1 2D Shallow Water Equations

The hydrodynamic model characterizes surface flow through the hyperbolic 2D SWE system, which is based on the mass and momentum conservation (Cunge et al., 1989):

90 
$$\frac{\partial h}{\partial t} + \nabla(h\mathbf{v}) = r - i + e + S_w$$
 (1)

$$\frac{\partial hu}{\partial t} + \frac{\partial}{\partial x} \left( u^2 + \frac{1}{2} gh^2 \right) + \frac{\partial (huv)}{\partial y} = -ghS_{fx} - gh\frac{\partial z_b}{\partial x} \tag{2}$$

$$\frac{\partial hv}{\partial t} + \frac{\partial}{\partial y} \left( v^2 + \frac{1}{2}gh^2 \right) + \frac{\partial (huv)}{\partial x} = -ghS_{fy} - gh\frac{\partial z_b}{\partial y} \tag{3}$$

where h is the water depth [L],  $\mathbf{v} = (u,v)$  is the velocity flow vector  $[LT^{-1}]$ , r is the precipitation rate  $[LT^{-1}]$ , i is the infiltration rate  $[LT^{-1}]$ , e is the evaporation rate  $[LT^{-1}]$  and  $S_w$  denotes the flow rate  $[LT^{-1}]$  provided by sources or sinks. The gravitational acceleration is denoted by  $g[LT^{-2}]$ ,  $z_b$  is the bed elevation [L], and  $S_{fx}$  and  $S_{fy}$  are the friction slope terms, defined as:

$$S_{fx} = \frac{n^2 u \sqrt{u^2 + v^2}}{h^{4/3}}, \quad S_{fy} = \frac{n^2 v \sqrt{u^2 + v^2}}{h^{4/3}}$$

$$\tag{4}$$

where n is the Manning's roughness coefficient  $[TL^{-1/3}]$  (Arcement and Schneider, 1984).

### 2.2 Particle transport equations

In a Lagrangian model of material transport, material elements are entrained and transported based on hydrodynamic forces, computed with appropriate coefficients or following a kinematic approach. In this study, the transported particles act as passive tracers: they have negligible volume and mass, do not interact with each other, and do not affect the flow field.

In the absence of turbulence, particle transport is driven by advection, and the particle trajectories are governed by the following system of equations that defines the position of each particle  $\mathbf{x}_p = (x_p, y_p, z_p)$ :

$$\begin{cases}
\frac{dx_p}{dt} = u(\mathbf{x}_p) \\
\frac{dy_p}{dt} = v(\mathbf{x}_p) \\
z_p = h(\mathbf{x}_p) + z_b(\mathbf{x}_p)
\end{cases} \tag{5}$$

As observed, the vertical position is unaffected by the flow velocity due to the depth-averaged SWE approximation, which neglects the vertical velocity component. Consequently, the particle is assumed to reside at the free surface, computed as the sum

of the water depth h and the bed elevation z<sub>b</sub>. This assignment is not derived from the governing equations but serves primarily to maintain numerical robustness and for visualization purposes. Since the hydrodynamic model is vertically averaged, and thus does not resolve vertical flow structure, particles transported by it do not possess a true vertical coordinate in the physical sense. However, assigning them a position at the free surface ensures consistency with the surface flow and avoids issues arising from irregular bathymetry. Notably, this choice prevents numerical artifacts, such as particles unrealistically crossing obstacles or walls due to inconsistent vertical velocities. Moreover, because the advection velocity is evaluated at the horizontal location of each particle, aligning all particles to the free surface provides a coherent reference for computing motion in the horizontal plane.

Turbulence effects in a flow can be significant and, by modifying the velocity field, also influence particle transport (Merritt and Wohl, 2002). Therefore, if turbulence is not incorporated into the flow velocity, as it is in the hydrodynamic model of the SERGHEI framework (Caviedes-Voullième et al., 2023), it can be included in the particle velocity, as follows:

$$\begin{cases}
\frac{dx_p}{dt} = u(\mathbf{x}_p) + u_{\text{disp}}(\mathbf{x}_p) \\
\frac{dy_p}{dt} = v(\mathbf{x}_p) + v_{\text{disp}}(\mathbf{x}_p) \\
z_p = h(\mathbf{x}_p) + z_b(\mathbf{x}_p)
\end{cases} (6)$$

where  $\mathbf{v}_{\text{disp}} = (u_{\text{disp}}, v_{\text{disp}}, 0)$  is the velocity induced by the dispersion, where it is considered a null dispersion velocity in the vertical coordinate because of the lack of the advection velocity in the vertical. In this study, a random-walk model is employed to simulate dispersion because of its simplicity and its great balance between computational efficiency and accuracy (Jalón-Rojas et al., 2019). This model proposes the  $\mathbf{v}_{\text{disp}}$  as a function of the diffusivity coefficients  $K_{hx}$  and  $K_{hy}$  [ $L^2T^{-1}$ ] in the x- and y-coordinates, respectively. In contrast to imposing constant values based on empirical results (Jalón-Rojas et al., 2019; Peeters and Hofmann, 2015) that depends on the size and shape of the object, the diffusivity coefficients  $K_{hx}$  and  $K_{hy}$  are here derived from a velocity-dependent expression. Therefore, the standard anisotropic diffusion model (Morales-Hernández et al., 2019) is utilized to determine the diffusivity variables, with the following expressions:

30 
$$K_{hx} = \epsilon_L |u^*| h(x_p, y_p), \quad K_{hy} = \epsilon_T |u^*| h(x_p, y_p)$$
 (7)

where  $\epsilon_L$ , and  $\epsilon_T$  are the longitudinal and transversal dispersion coefficients  $[L^2T^{-1}]$  (Rutherford, 1994), respectively; and  $|u^*|$  is the friction velocity:

$$|u^*| = n\sqrt{g\frac{v_x^2 + v_y^2}{(h(x_p, y_p))^{1/3}}}$$
(8)

#### 3 Numerical schemes

120

The equations (1), (2) and (3) are solved using an explicit upwind finite volume scheme, which is based on the Augmented Roe Riemann solver. Further details about the numerical scheme can be found in (Echeverribar et al., 2019; Murillo and García-

Navarro, 2010; Morales-Hernández et al., 2013), together with the HPC implementation within SERGHEI (Caviedes-Voullième et al., 2023). In this paper, several numerical schemes are explored to solve the particle equations systems.

## 3.1 Numerical scheme for the LPT model

To obtain the temporal and spatial evolution of the particles, it is necessary to discretize the system of equations (5) when turbulence is not considered, or system (6) when it is included. In passive particle transport, particles are considered to have negligible mass and volume, ensuring no interaction with each other or the flow, thereby maintaining unaltered flow properties such as density or viscosity. The choice of discretization method depends on the required accuracy. In this study, two methods are utilized to discretize the equations governing particle trajectories: the forward Euler method and the Runge-Kutta method.

## 145 3.1.1 Explicit forward Euler method

150

The temporal evolution of particle positions is obtained by considering the particles as mathematical points denoted by **p**, which are advected in space **x** following a velocity field **v**. In this model, dispersion is not calculated within the hydrodynamic module, potentially leading to unrealistic particle transport if this phenomenon is not considered (Merritt and Wohl, 2002). Although the flow still has no turbulence model, the dispersive effects of turbulence are modelled on the transport of particles. These effects are added using a random-walk model, which can generate sufficient dispersion to introduce turbulent motion (Rutherford, 1994). Including the turbulence terms by using a random-walk model, the discretization of the system of equations (6) is as follows:

$$\begin{cases} x_p^{n+1} = x_p^n + \Delta t_p^n u_i^n + \Delta t_p^n u_{\text{disp},p}^n = x_p^n + \Delta t_p^n u_i^n + R_{p,x}^n \left( 2\sigma^{-1} K_{hx,i}^n \Delta t_p^n \right)^{1/2} \\ y_p^{n+1} = y_p^n + \Delta t_p^n v_i^n + \Delta t_p^n v_{\text{disp},p}^n = y_p^n + \Delta t_p^n v_i^n + R_{p,y}^n \left( 2\sigma^{-1} K_{hy,i}^n \Delta t_p^n \right)^{1/2} \\ z_p^{n+1} = h_i^n + z_{b,i} \end{cases}$$
(9)

with the dispersion velocities defined by:

$$u_{\mathrm{disp},p}^{n} = \sqrt{\frac{2K_{hx,i}^{n}}{\sigma\Delta t_{n}^{n}}}, \quad v_{\mathrm{disp},p}^{n} = \sqrt{\frac{2K_{hy,i}^{n}}{\sigma\Delta t_{n}^{n}}}$$
(10)

where the Eulerian velocity field  $\mathbf{v}_i^n$  is the velocity field in cell i at time step n, in which the particle p is located, i.e., contains the point  $\mathbf{x_p}^n = (x_p^n, y_p^n, z_p^n)$ ,  $h_i^n$  is the water depth in the cell i at time step n,  $z_{b,i}$  is the bottom elevation of the cell i, and  $\Delta t_p^n$  is the particle time step.  $R_{p,x}^n$  and  $R_{p,y}^n$  are random numbers, which follow a uniform distribution with mean 0 and standard deviation  $\sigma = 1$ .  $K_{hx,i}^n$  and  $K_{hy,i}^n$  are the horizontal diffusivity in  $\hat{x}$  and  $\hat{y}$  components, respectively, as defined by expression (7).

#### 160 3.1.2 Runge-Kutta method

The primary limitation of the explicit Euler method is its first-order accuracy, which may result in significant deviations from reality unless the time step for the Lagrangian model is sufficiently small (Bennett and Clites, 1987). Therefore, a higher-order

approximation is necessary to capture the spatial variation of the velocity field when a large time step is computed by the numerical scheme. For this reason, a fourth-order Runge-Kutta method is presented for the advection formulation:

$$\mathbf{165} \quad \mathbf{x_p}^{n+1} = \mathbf{x_p}^n + \frac{\Delta t_p^n}{6} \left( \mathbf{v} \left( \mathbf{x_p}, t^n \right) + 2\mathbf{v} \left( \mathbf{x_p} + \frac{1}{2} a, t^{n+1/2} \right) + 2\mathbf{v} \left( \mathbf{x_p} + \frac{1}{2} b, t^{n+1/2} \right) + \mathbf{v} \left( \mathbf{x_p} + c, t^{n+1} \right) \right)$$

$$(11)$$

where:

185

$$a = \Delta t_p^n \mathbf{v} \left( \mathbf{x_p}, t^n \right), \quad b = \Delta t_p^n \mathbf{v} \left( \mathbf{x_p} + \frac{1}{2} a, t^{n+1/2} \right), \quad c = \Delta t_p^n \mathbf{v} \left( \mathbf{x_p} + \frac{1}{2} b, t^{n+1/2} \right)$$

$$(12)$$

This method requires the velocity at intermediate times  $(t^{n+1/2})$ , which is not directly calculated by the numerical scheme. Therefore, a fourth-order interpolation is used to obtain it:

170 
$$\mathbf{v}^{n+1/2}(\mathbf{x}_{\mathbf{p}}) = \frac{5}{16}\mathbf{v}^{n+1}(\mathbf{x}_{\mathbf{p}}) + \frac{15}{16}\mathbf{v}^{n}(\mathbf{x}_{\mathbf{p}}) - \frac{5}{16}\mathbf{v}^{n-1}(\mathbf{x}_{\mathbf{p}}) + \frac{1}{16}\mathbf{v}^{n-2}(\mathbf{x}_{\mathbf{p}})$$
 (13)

Comparing the equations of both methods, it is evident that the higher accuracy of the Runge-Kutta method entails a higher computational cost per time step than the explicit Euler method, as the Runge-Kutta method requires the velocity vector at four different times and the calculation of additional terms to determine the trajectory evolution.

### 3.2 Trajectory algorithm

The transport of the particle is governed by the advection. The trajectory can be determined directly from the initial cell to the final cell (García-Martínez and Flores-Tovar, 1999; Bennett and Clites, 1987) (see red dashed trajectory in Figure 1), or the particle can be moved cell by cell (see green dotted trajectory in Figure 1). The latter approach is compulsory for addressing potential dry-wet interface issues and for preventing particles from traversing obstacles, as shown in Figure 1b.

Thus, the algorithm that loops over the particles must be carefully designed to ensure that each particle traverses each computational cell discretely, dynamically adopting the local velocity and updating its position based on the discrete flow velocities within each cell.

For that, let us consider a single particle p defined by its position  $\mathbf{x}_{\mathbf{p}}$  in the same position of the point  $\mathbf{p}=(p_x,p_y)$ , which is inside of cell i, with some velocity  $\mathbf{v}_i$ . Let d be the distance between the particle and the point at which the particle would leave the cell when following the velocity within the cell, as shown in Figure 2a. In this figure,  $\mathbf{A}=(A_x,A_y)$  and  $\mathbf{q}=(q_x,q_y)$  are the lower left and right corners (respectively) of the cell, b is the distance between  $\mathbf{q}$  and the intersection edge point, and  $\hat{\mathbf{e}}$  is the unit vector of the Cartesian y-axis. The distance d is the minimum distance in the direction of the velocity vector  $\mathbf{v}$  from the point  $\mathbf{p}$  until it crosses an edge that separates one computational cell from the next. This distance is measured along the line defined by the unit vector  $\hat{\mathbf{v}}$ .

In order to guarantee that the particle is not advected beyond cell i with the velocity  $\mathbf{v}_i$ , it follows that a geometric restriction for the time step  $\Delta t_p$  of particle p is given by:

$$\Delta t_p \le \frac{d}{||\mathbf{v}_i||} \tag{14}$$

**Figure 1.** The two possible strategies for the trajectory of particles in absence of obstacles (a) and the same strategies when there is an obstacle (gray cell) (b).

Figure 2. (a) Representation of a point p (particle position) in a generic cell and (b) a generic representation of the time steps for 2D model and particles.

This is fundamentally a CFL condition for the particle advection. An additional restriction applies for  $\Delta t_p$  related to the hydrodynamic time step size  $\Delta t$ :

$$\Delta t_p \le \Delta t \tag{15}$$

ensuring the particle model cannot advance further in time than the hydrodynamic field. When  $\Delta t_p 

Figure 3. Three possible situations for a particle: traveling through 1 cell (a), 2 cells (b), or 3 cells (c).

#### The value of d is calculated as follows:

1. Given a line with with direction  $\hat{\mathbf{v}} = \mathbf{v} ||\mathbf{v}||^{-1}$  passing through particle point  $\mathbf{p}$ , find its intersection with the lines defining the edges, and compute the distance d from  $\mathbf{p}$  to the 4 intersection points: north, south, east and west intersections (see

Figure 2):

$$\mathbf{p} + d\hat{\mathbf{v}} = \mathbf{q} + b\hat{e} \tag{18}$$

215 Considering only the *x*-coordinate:

$$p_x + d\hat{v_x} = q_x \tag{19}$$

and therefore, if  $\hat{v_x}$  is not null:

$$d = \frac{q_x - p_x}{\hat{v_x}} = \frac{A_x + \Delta x - p_x}{\hat{v_x}} \tag{20}$$

where  $\Delta x$  is the cell length in the x-coordinate, which, being a square cell, is equivalent to the length in the y-coordinate. For cases where  $\hat{v_x}$  is null or negative, the distance d calculated using (20) does not yield the correct intersection. For this reason, three additional distances d are calculated by employing a procedure analogous to that used for deriving (20):

$$d_2 = \frac{A_x - p_x}{\hat{v_x}} \qquad \text{if and only if} \quad \hat{v_x} \neq 0 \tag{21}$$

$$d_3 = \frac{A_y + \Delta x - p_y}{\hat{v_y}} \qquad \text{if and only if} \quad \hat{v_y} \neq 0$$
 (22)

225

235

220

$$d_4 = \frac{A_y - p_y}{\hat{v_y}} \qquad \text{if and only if} \quad \hat{v_y} \neq 0 \tag{23}$$

- 2. Discard negative distances as these are antiparallel to the velocity vector.
- 3. The minimum positive distance d corresponds to the correct intersection.

Having computed d, it is possible to evaluate the particle sub-cycling time step (17), and with such time step advect the particle with (6). This algorithm is used in both the Euler and Runge-Kutta methods.

Thus, the temporal evolution of the x- and y-coordinates of particles are obtained by the proposed algorithm. With respect to the z coordinate, assuming a passive particle transport with negligible mass and volume for the particles, the value of the water surface level of the cell where the particle is located is imposed, so it is floating in it. This approach and the use of the algorithm automatically resolve issues such as: (i) the wet/dry problem, provided the numerical method accurately addresses these conditions for the flow; and (ii) wet/wet problems involving topographical changes, where a particle floating at an arbitrary vertical position within a cell may encounter an adjacent cell with a higher elevation, preventing its movement into that cell despite both cells containing water. This ensures that the particle's movement is physically feasible and adheres to the elevation constraints of the cells.

Introducing a turbulent dispersion term can cause issues in wet-dry cases, as a particle could find itself in a wet cell due to advection and then reach a dry cell when the dispersion term is applied. Therefore, a restriction is imposed: if the destination cell (determined by adding the dispersion term) is dry, the dispersion term is not applied. This restriction is implemented for several reasons:

- It is physically inconsistent for a particle with negligible mass to enter a dry cell. The underlying principle is that particles with significant mass could use inertia to traverse dry regions. Therefore, this restriction specifically applies to massless or negligible-mass particles.
- The dispersion term can relocate the particle horizontally across the surface, and if this displacement places it in a dry cell with a higher elevation than its previous location, the particle undergoes an unphysical "jump" in its trajectory. This arises because dispersion terms depend on friction velocity without properly accounting for flow direction constraints.

## 3.3 Temporal integration modes

245

Depending on the desired accuracy of the results, different time intervals are proposed to update the evolution of the particles. When the particle advection is computed on every time step of the hydrodynamic solver (from n to n+1), this configuration is referred to as the online mode. In contrast, when particle positions are updated every nIter hydrodynamic time steps (from n to n+nIter), the configuration corresponds to the offline mode. The online mode is expected to yield higher accuracy than the offline mode, but potentially with a higher computational cost due to more frequent advection updates. The explicit Euler method is unsuitable for offline mode due to its limited accuracy, and is therefore applied exclusively in the online configuration. Conversely, the Runge–Kutta method demonstrates sufficient accuracy to be employed in both online and offline modes.

## 3.4 Implementation in SERGHEI framework

The LPT model has been implemented in the SERGHEI framework, as a one of the several modules that have been implemented in this framework (see Figure 4). The SERGHEI framework is a modular and open-source simulation system designed to target performance portability across HPC (High-Performance Computing) platforms and standard workstations (Caviedes-Voullième et al., 2023). It focuses on future-proofing and sustainability while consolidating over a decade of mature numerical techniques for shallow water modeling. SERGHEI facilitates a modular, HPC-ready community model, enabling broader collaboration. The framework moves beyond classical fluvial flooding and hydrograph generation to address applications related to landscape function, transport, and integration into Earth System Models (ESM).

The LPT model has been developed along the same lines as the other modules in SERGHEI. In this way, this model has been implemented in C++ and using the *Kokkos* performance-portability layer, which allows the simulation of the model using any GPU architecture. As can be seen in Figure 5, the LPT model is updated after updating the hydrodynamic model, as it needs the updated information of the conserved variables. The LPT model is currently being implemented to support distributed computations both on multiple CPU-nodes (note that shared memory CPU parallelisation is achieved via OpenMP) and on multi-GPU systems, following the approach used in other SERGHEI modules.

Figure 4. Conceptual diagram for the SERGHEI modules and their implementation status.

**Figure 5.** Schematic of the numerical calculation using the LPT module.

#### 4 Results

The validation of the LPT model is conducted through the simulation of several test cases with increasing complexity, assessing both accuracy and computational efficiency. The simulations are run using a GPU NVIDIA GeForce RTX 3060.

## 4.1 Steady circular vortex case

Initially, a steady circular vortex test case is simulated to validate the model's accuracy. The vortex has a diameter of 10 meters 275 and is centered in the middle of the domain. Its velocity field increases radially from zero at the center to a maximum of 100 m s<sup>-1</sup> at the outer edge. A steady-state configuration is assumed, meaning that particles released into the flow should maintain a constant radius throughout their trajectories (Finaud-Guyot et al., 2023; García-Martínez and Flores-Tovar, 1999). To ensure reproducibility, the particle is released at a distance of 10 meters from the vortex center. The circular vortex is simulated with 280 varying cell sizes, ensuring the particle completed four full revolutions for consistency in error measurement. The vortex radius is consistently set at R = 100 m, with a water depth of h = 0.1 m and no roughness (n = 0). For this test, the hydrodynamic solver is turned off, so that the Eulerian flow fields (water depth and velocity) remained constant and are not evolved over time, also resulting in a constant time step  $\Delta t$  throughout the simulation. Turbulent diffusion terms are excluded from the particle trajectory equations because this is an analytical case, and the objective is to guarantee correctness and robustness 285 of the advection. The Runge-Kutta offline method is updated every five hydrodynamic time steps, a choice made to balance computational efficiency and accuracy. Preliminary tests with shorter update intervals (2–3 time steps) yielded only marginal cost savings and limited improvement in accuracy, so the 5-step update was selected to better illustrate this trade-off.

The errors for each grid resolution using different numerical configurations are shown in Figure 6. The  $L_1$ -norm for the error at each time step is defined by:

$$290 \quad L_1(t^n) = |\mathbf{x}_{\mathbf{p}}^n - \hat{\mathbf{x}}| \tag{24}$$

where  $\mathbf{x}_{\mathbf{p}}^{n}$  is the particle position at the time  $t^{n}$  and  $\hat{\mathbf{x}}$  is the analytical position. The  $L_{1}$ -norm integrated in time is the Mean Absolute Error (MAE), which is obtained using:

$$MAE = \frac{1}{N} \sum_{n=0}^{N} L_1(t^n)$$
 (25)

where N is the number of time measurements.

Additionally, the  $L_2$ -norm at each time step is defined as:

$$L_2(t^n) = (\mathbf{x}_{\mathbf{p}}^n - \hat{\mathbf{x}})^2 \tag{26}$$

while the Root Mean Squared Error (RMSE) is computed as:

$$RMSE = \sqrt{\frac{1}{N} \sum_{n=0}^{N} L_2(t^n)}$$
 (27)

Figure 6 shows that, under steady flow, the error of Runge-Kutta method is similar to the Euler method. The algorithm implemented provides a reduced error for the Euler method because of the calculus of the trajectory in each cell (see Figure 1) and in each time step. Moreover, the error of all methods depends strongly on the discretization of the domain, with lower errors for smaller cell sizes. Figure 6 also shows the error reduction in the same order as the Euler method as the cell size is reduced (order 1). However, for the Runge-Kutta method (order 4), the error is not observed to reduce with that order. This is mainly because it is a steady case and the value of the time step is fixed.

**Figure 6.** Errors simulating the steady circular vortex case using different numerical methods, shown in logarithmic scales: MAE (a) and RMSE (b). A first order convergence slope is included for reference.

#### 305 4.2 Transient test case

310

300

Following the accuracy assessment, an analysis of computational efficiency is conducted using a well-known test case (see Figure 7a), previously simulated and compared against measurements using the SERGHEI framework (Caviedes-Voullième et al., 2023). The details of the experimental setup are described in (Soares-Frazão and Zech, 2008), where the domain dimensions are L=36 m in length, B=3.6 m in width. As the initial condition, an initial water depth of 0.4 m in the reservoir region and 0 m in the right region are imposed, modifying the value of the initial condition in the right region compared to the original experiment to check the accuracy of the model when wet-dry situations occur. The comparative analysis is based on increasing the number of particles to evaluate the model computational cost for each configuration. Moreover, this case is used as a validation of the model's ability to correctly resolve wet-dry situations. Reflective boundary conditions are applied everywhere, maintaining a constant number of particles throughout the simulation. The Runge-Kutta offline method was updated using different time periods  $(5\Delta t, 10\Delta t, 20\Delta t$  and  $50\Delta t$ , where  $\Delta t$  is the hydrodynamic time step) to analyze the influence of this variable in the temporal evolution of the particles position.

**Table 1.** Computational cost for different number of particles using the numerical methods implemented and the increase ratio between simulation with particles and without particles, for the dam break test case.

| Numerical method                | Number of particles | Computational cost (s) | Increase ratio |
|---------------------------------|---------------------|------------------------|----------------|
| Euler                           | $10^4$              | 13.15                  | 1.08           |
| Euler                           | $10^{5}$            | 21.22                  | 1.75           |
| Euler                           | $10^{6}$            | 66.59                  | 5.48           |
| Runge-Kutta                     | $10^4$              | 30.00                  | 2.47           |
| Runge-Kutta                     | $10^{5}$            | 121.93                 | 10.03          |
| Runge-Kutta                     | $10^{6}$            | 1043.56                | 85.89          |
| Runge-Kutta Offline $5\Delta t$ | $10^4$              | 18.89                  | 1.55           |
| Runge-Kutta Offline $5\Delta t$ | $10^{5}$            | 60.93                  | 5.01           |
| Runge-Kutta Offline $5\Delta t$ | $10^{6}$            | 474.24                 | 39.03          |

The computational costs for the different configurations with different particle numbers are shown in Table 1, where only the Runge-Kutta offline computational costs updated every 5 time steps are shown because it is the less computationally efficient case for the simulated Runge-Kutta offline combinations. The computational time for this simulation without particles was 12.15 seconds, and an overhead ratio relative to the hydrodynamic-only run is computed for each case. The forward Euler method consistently demonstrated shorter computational times compared to other methods across all configurations. Additionally, four states during the simulation are depicted in Figure 7, illustrating realistic particle trajectories simulated with the Euler scheme without traversing dry cells or climbing over buildings. However, when the Runge-Kutta offline is updated using long time periods, the temporal evolution of particles position can be illogical, as shown in Figure 8, where it can be seen some particles climbing over buildings. Finally, the differences between the methods' results are shown in Figure 9, where the  $L_2$ -norm per particle is derived from the differences between models, taking the Runge Kutta online method as the basis for comparison. The error between the Euler and Runge-Kutta method is lower than the errors between the Runge-Kutta and Runge-Kutta offline methods, indicating that the Euler method provides results more similar to the Runge-Kutta method while maintaining higher computational efficiency. In addition, as the update time for the offline Runge-Kutta method increases, the difference between the online Runge-Kutta and the offline Runge-Kutta increases too, due to the fact that being a transient case, not updating frequently can lead to a high error.

Figure 7. Initial condition of the transient test case (a). States at t = 5s (b), at t = 10s (c), at t = 15s (d), and at t = 20s (e) for the Euler method with  $10^5$  particles in the dam break case.

## 4.3 Channel with cavities

A longitudinal channel with symmetrically shaped cavities was simulated, using a mesh with a resolution of 2x2 cm (see Figure 10a). The objective is to check whether a symmetrical distribution of particles is achieved across the upper and lower cavities (Vallés et al., 2023). 10<sup>5</sup> particles were simulated with an initial distribution as shown in Figure 10a. Given the best compromise

Figure 8. State with particles climbing over buildings for the Runge-Kutta offline simulation with time step  $50\Delta t$  (a), and zoom of the building area (b).

Figure 9. Temporal evolution of the  $L_2$ -norm per particle obtained from the difference between methods results.

between accuracy and computational efficiency in the last test cases, the Euler method was only employed for this simulation. The temporal evolution of the particles position with and without turbulence was compared (see Figure 10), with the evolution of particle numbers in each pair of upper and lower cavities depicted in Figure 11. The presence of turbulence resulted in greater

dispersion in particle distribution and more realistic trajectories compared to simulations without turbulence. This is further evidenced in Figure 11, where the number of particles within a cavity is higher when turbulence is considered. The symmetry between the upper and lower cavities (see Figure 11), regardless of whether turbulence is added or not, provides a further argument for the accuracy of the Lagrangian model.

An important observation is that with coarser grid resolutions, the velocity field may not be sufficiently resolved to drive particles into the cavities. To explore this, the same simulation was repeated with and without dispersion using a different grid resolution: 20 cm. As shown in Figures 12 and 13, the coarser mesh failed to guide particles into several cavities, with some receiving no particles at all. In contrast, the finer resolution (see Figure 11) captured the expected cavity entry more accurately. This highlights a key limitation of coarse Eulerian velocity fields, which can be partially mitigated by including the random-walk model: it enhances subgrid-scale transport and improves the physical plausibility of particle trajectories. Although this test does not constitute formal model verification, the observed symmetry in particle distributions—particularly under the influence of turbulence—provides soft validation that the Lagrangian model behaves in a consistent and physically reasonable way, without introducing spurious asymmetric artifacts.

## 4.4 Example application: surface runoff in the Arnás catchment

The implemented Lagrangian model may have applications in ecohydrology, such as seed, pollutant, or nutrient transport. Consequently, a case involving a precipitation event over the Arnás catchment (see Figure 14a) was simulated to show the potential application of this model. This catchment is located in the Borau valley (Northern Spanish Pyrenees). In the last decades, this region has been an area of exhaustive study of different hydrological processes (Lana-Renault et al., 2007; García-Ruiz et al., 2005). Physical processes such as rainfall, infiltration, runoff generation, and gully formation have been simulated in detail using various methods (López-Barrera et al., 2011; Fernández-Pato et al., 2016; Vallés et al., 2024). The primary objective of simulating this case was to assess its potential utility in environmental applications. Information derived from conservative tracers and field experiments measuring isotopes in water has been widely used in hydrology to characterize hydrological pathways and connectivity, with ideas such as transit or residence times, and travel time distributions (Benettin et al., 2022). Here, a minimal example is presented to demonstrate how analogous ideas can be investigated within the proposed computational modelling framework.

The catchment was discretized at 5m resolution, for a total of  $1.1 \times 10^5$  computational cells.  $10^4$  number of particles were randomly placed in the domain as initial condition for the LPT model, and the catchment was initially dry for the shallow water solver. As in the previous test case, the Euler method was exclusively used for the simulation of the Arnás case, based on the optimal balance between accuracy and computational efficiency observed in the initial test cases. Two real precipitation cases (see Figure 14b and 14c) were simulated to analyze the particle trajectory behaviour under different hyetographs. In both events, particles are mostly transported through the main gullies. The runtimes for the simulations of events 1 and 2 were 317.75s and 177.02s respectively, corresponding to runtime increase ratios of  $2.39 \times$  and  $1.10 \times$  compared to the same simulations without particles.

Figure 10. Initial condition for the cavities test case with  $10^5$  particles with the numeration of the polygons (a). States with zoom at t = 50s without turbulence (b) and with turbulence (c), and at t = 100s without turbulence (d) and with turbulence (e).

Additionally, the total travel distance per particle, the particle travel time (travel time), and the time during which the particle was at rest (stop time) are shown in Figures 15 and 16 for each precipitation event. These figures provide a statistical overview of particle dynamics during the runoff events. Physically, these metrics offer insights into the hydrological processes at play. The travel distance (displayed in Figure 14a) reflects the efficiency of the drainage network of the basin to transport the debris, which can be seen in Figures 15a and 15b, where most particles travel a distance very similar to the initial one. Moreover, Figure 15a shows that a subset of particles from event 2 travels much farther than the rest (around 10 km). This is due to the properties of this precipitation event, which features less intense but longer rainfall than event 1. This extended duration causes particles to travel for longer periods (with shorter stop times), as also observed in Figure 15d. Furthermore, this histogram reveals regions where particles tend to accumulate or remain at rest for extended periods, typically due to reduced flow velocities or topographical

375

**Figure 11.** Temporal evolution of the number of particles inside of the different cavities (cavities 1 and 5 in Figure (**a**), cavities 2 and 6 in Figure (**b**), cavities 3 and 7 in Figure (**c**), and cavities 4 and 8 in Figure (**d**)).

barriers, which may indicate zones of storage or potential sediment deposition. However, as shown in Figure 16, the set of particles with high travel distances does not influence the last hours of the temporal evolution of the total travel distance (see Figure 16) because in both events the majority of the particles go out the domain. However, the difference between the events is higher at the beginning of the case, having a big difference between both hyetographs in the first hours. In event 1, the longer travel times observed for most particles in Figure 15c, where particles tend to remain approximately at rest for extended periods, suggest intermittent flow conditions, possibly due to intermittent rainfall, leading to temporary particle deposition. In contrast, event 2 exhibits more uniform movement and shorter stop times, likely due to less intermittent rainfall that fosters persistent

Figure 12. States with zoom at t = 50s without turbulence (a) and with turbulence (b), and at t = 100s without turbulence (c) and with turbulence (d) for a resolution of 20 cm.

particle transport. Thus, these histograms not only quantify particle behavior but also provide insights into the underlying hydrological and geomorphological processes shaping the catchment's response to precipitation events.

#### 5 Conclusions

In this work, a Lagrangian model for particle transport implemented within the SERGHEI framework is presented. This model utilizes an algorithm that leverages information from each cell occupied by the particles and is solved using three numerical schemes: an online first-order Euler method, an online fourth-order Runge-Kutta (RK4) method, and an offline RK4 method.

The convergence and accuracy of these solvers were first assessed using a steady circular vortex benchmark case. All methods yielded comparable error metrics and demonstrated proper grid convergence behavior. In particular, the Euler and RK4 online schemes produced nearly identical results under steady conditions, highlighting the accuracy of the Euler method despite its lower order.

Next, a transient test case without dispersive terms was examined to evaluate computational efficiency and the handling of dry-wet interfaces. Results showed that the Euler method was the most computationally efficient, with relatively small differences in accuracy compared to the RK4 schemes. Furthermore, as shown in Figure 9, the offline RK4 method produced incorrect trajectories due to its inability to account for dry cells during interpolation. The high accuracy of the Euler scheme is attributed to the sufficiently small time steps used in the simulations. While RK4 online may offer slightly improved accuracy in transient flows with fewer particles, its computational cost increases significantly with particle count—up to 17 times more than Euler in cases involving  $10^5$  particles or more. Given that this model is designed for large-scale applications, where millions of particles may be simulated, the Euler method offers the best trade-off between accuracy and efficiency, making it the preferred scheme for practical use.

**Figure 13.** Temporal evolution of the number of particles inside of the different cavities (cavities 1 and 5 in Figure (a), cavities 2 and 6 in Figure (b), cavities 3 and 7 in Figure (c), and cavities 4 and 8 in Figure (d)) for a resolution of 20 cm.

To investigate the impact of turbulence, a channel with multiple cavities both with and without turbulence was simulated. The results illustrate that incorporating turbulence significantly alters particle trajectories, resulting in more realistic transport dynamics. However, as it is shown in previous work (Vallés et al., 2024), the addition of the turbulence implies a relevant increasing of the computational cost. Additionally, a comparison with a coarser grid resolution demonstrated that insufficient spatial resolution in the Eulerian velocity field may prevent particles from entering cavity regions. This limitation can be partially mitigated by including the random-walk dispersion model, which enhances subgrid-scale transport and contributes to more physically consistent trajectories. The observed symmetry in particle distributions, especially under turbulent conditions, provides further support for the model's qualitative reliability.

**Figure 14.** 3D representation of the Arnás catchment, including the travel distance and the initial distance (a), and hyetographs of event 1 (b) and event 2 (c) for the Arnás case.

Finally, the model was applied to a realistic case study of the Arnás catchment. The output generated by the model offers significant insights for environmental applications. The results elucidate the mechanisms of particle transport during precipitation events and identify the primary gullies that facilitate this transport. Furthermore, this information provides critical insights into the dynamics of particle movement, revealing distinct patterns in both movement and stop times. These patterns reflect the influence of varying precipitation intensities and can help on identifying sediment mobilization and deposition processes within the catchment. This analysis underscores the potential of such models to enhance our understanding of different multiscale complex phenomena that take places in a catchment.

Figure 15. Histograms of: (a) the travel distance, (b) the travel distance normalised by the initial distance between the particle and the boundary condition position for both precipitation events, (c) the travel time, and (d) the stop time.

In summary, the Euler method delivers a strong balance between computational efficiency and numerical accuracy, particularly in large-scale simulations where performance is critical. The Lagrangian particle transport (LPT) model developed within the SERGHEI framework effectively reproduces realistic particle trajectories and can account for complex interactions such as dry-wet interfaces and turbulence. Future work will focus on optimizing the model to further reduce computational costs and implementing multi-GPU simulations to leverage the capabilities of the SERGHEI hydrodynamic model. Moreover, several enhancements will be incorporated into the LPT module to increase the realism of particle trajectories. These improvements will

Figure 16. Temporal evolution of the total travel distance for both events.

enable representation of pollutant transport (e.g., microplastics) and biological dispersal (e.g., seeds). Additionally, incorporating particle mass, volume, and inertia will allow modeling of vertical movement, deposition processes, and macroscopic objects such as wooden logs or urban debris in floods. However, accurate simulation of these transport phenomena requires development and validation of specific physical processes that are currently beyond the scope of the present model.

Code and data availability. SERGHEI is available through GitLab, at https://gitlab.com/serghei-model/serghei, under a 3-clause BSD license. SERGHEI v2.1.0 was tagged as the first release of the LPT module at the time of submission of this paper. Static code for v2.1.0 can be found at https://doi.org/10.5281/zenodo.14871005 (Vallés et al., 2025b). This release does not include the Runge-Kutta solver. Static code for the experimental solution with Runge-Kutta integrators can be found at https://doi.org/10.5281/zenodo.14870918 (Vallés et al., 2025a). A repository containing test cases is available https://gitlab.com/serghei-model/serghei-tests/lpt.

Author contributions. PV: Investigation, Methodology, Software, Validation, Writing - Original Draft Preparation. DCV: Supervision, Conceptualization, Methodology, Software, Formal analysis, Data Curation, Writing - Review and Editing. MMH: Supervision, Funding acquisition, Conceptualization, Methodology, Formal analysis, Writing - Review and Editing. VR: Supervision, Funding acquisition, Writing - Review and Editing. PGN: Funding acquisition, Writing - Review and Editing.

Competing interests. No competing interests exist.

Acknowledgements. Pablo Vallés is funded by the UPPA-UNIZAR Research Grant PI-PRD/2022-03 and he was funded by ERASMUS+ KA103 "IBERUS+" 2021-1-ES01-KA130-HED-000004265. This work was supported by project PID2022-137334NB-I00 funded by MCIN/AEI/10.13039/501100011033 and by ERDF/EU. This work was partially funded by the Government of Aragón through the research grant T32\_23R Fluid Dynamics Technologies and by the project JIUZ2023-IA-04 from UNIZAR.

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
