# Peer review of "SERGHEI v2.1: a Lagrangian Model for Passive Particle Transport using a 2D Shallow Water Model (SERGHEI-LPT)"

_EGUsphere, 2025_

## Referee Comment (RC1)

The paper presents a Lagrangian method to model the transport of no-mass particles, implemented in the SERGHEI framework. The introduction is easy to read and comprehensive, with up-to-date references. The algorithm is discretized with three different approaches, which are systematically tested to assess the computational cost and overall efficiency. The contribution is very interesting, well-structured and carefully organized.

As the "passive particles" are not clearly identified (they could be microplastics, seeds, pollutants…) the fact that the model only includes transport is fine, although further enhancement would be required to model specific dynamics (like deposition, sedimentation, entrapment) that depends upon specific particle types. This lack of specificity is totally fine but could be discussed in conclusions to better contextualize the potentialities of the proposed model.

My main concern is about a restriction included in the model, to avoid particles reaching a dry cell because of turbulence. This is fine, in general, but may prevent natural behavior of particles that can actually reach dry cells, if I well understood the implemented approach. This should be discussed in more detail, as stated in the specific comments.

Please, find below my specific comments and a few typos to be corrected.

**Specific comments**

Line 44: "are designed for coastal scenarios where challenging wet-dry transitions do not occur". This sentence appears counterintuitive. Generally, on the coast, waves give rise to wet-dry scenarios. Please, specify what you mean.

System of equations 9: Please specify that the system of equations (9) represents the discretization of the system of equations (6). Additionally, you could explicitly describe the connection between $v_{disp}$ in system (6) and the random-walk model in system (9).

Lines 188-190: The time step of the particle is not necessarily an exact divisor of the hydrodynamic time step, as shown also in Figure 2b. In Eq. (15), the particle time step is different from the one computed by Eq. (14)? If so, how is the index "m" imposed? And, please check Eq. (15) versus Eq. (16), because they seem inconsistent (if Eq. (15) is correct, the summation of the particle time step is equal to the hydrodynamic time step, so the hydrodynamic time step minus the summation should be equal to zero in Eq. 16).

Lines 227-228: This restriction appears quite strict. It appears to limit the effect of turbulence. Are the effect of such limitation discussed? Are the Authors planning to remove it in a later version of the code? Furthermore, is it possible for transported objects that reach a dry cell to stop there?

Figure 4: The logical connection between modules is not fully clear. The figure shows a "Lagrangian particle transport" module (LPT), while in the text it is referred to as "Lagrangian model". Using the same term in the text and in the figure would help the reader, also in Fig. 5. Finally, what do the Authors mean with "Lagrangian model for distributed computations" (lines 249-250)?

Figure 6 is not clear: are all the three errors normalized by the Euler error? Apparently not (the Euler error should be one), and this is in contrast with both the vertical axis in the figure and the figure caption. The strong dependency on the domain discretization is not clear, either. It appears clear from the MAE and RMSE written in the figures, but not for the graphs. Please, consider to modify this figure.

Figure 8: please consider changing the particles color or zooming in the image to make the particles more visible.

Caption of Figure 10: 10000 particles are reported. This appears in contrast with line 309, where it is written that the simulation was performed with 1000000. Please, check.

Line 344: Can the Authors clarify what they mean by "areas of stagnant transport"?

Line 348-349: Can the Authors clarify what they mean by "the higher frequency of particles compared to the event 2"?

Conclusions: Nothing is said about model's future developments. Are the Authors planning to include a strategy to account for particles deposition? This also depends on the type of particles that they are aiming to model (plastics, seeds…). The work would possibly benefit from a more critical analysis of the potential applications of the model.

**Typos**

Line 175: I guess there is a typo, as "respectively" is repeated twice.

Caption of Figure 3: "final position" seems unnecessary.

Line 266: It should be "$L_1$ norm", and not "$L_1$ Norm". The same for $L_2$ norm at line 270.

Line 283: "described" semms unnecessary.

Line 284: "modying" should be "modifying"?

Line 300: another way of presenting the $L$ norm is used. Please be consistent in the terminology (choose between $L$ Norm, $L$ norm or $L$-norm).

Line 332: "This" and not "these figure".

Line 339: Please, use the same term, "travelled distance" or "covered distance" for higher consistency.

Line 347: "the difference between the events is higher"

---

## Author Response (AR1)

Dear Editor,

Please find enclosed the files corresponding to the revised version of the manuscript titled SERGHEI v2.1: a Lagrangian Model for Passive Particle Transport using a 2D Shallow Water Model in SERGHEI (ref: egusphere-2025-722). The manuscript has been modified and improved according to the reviewers' suggestions. The specific ways in which their comments have been addressed are listed below. All changes in the manuscript have been marked in colour.

Best regards,

The Authors

**Reviewer 1**

The paper presents a Lagrangian method to model the transport of no-mass particles, implemented in the SERGHEI framework. The introduction is easy to read and comprehensive, with up-to-date references. The algorithm is discretized with three different approaches, which are systematically tested to assess the computational cost and overall efficiency. The contribution is very interesting, well-structured and carefully organized.

As the "passive particles" are not clearly identified (they could be microplastics, seeds, pollutants...) the fact that the model only includes transport is fine, although further enhancement would be required to model specific dynamics (like deposition, sedimentation, entrapment) that depends upon specific particle types. This lack of specificity is totally fine but could be discussed in conclusions to better contextualize the potentialities of the proposed model.

My main concern is about a restriction included in the model, to avoid particles reaching a dry cell because of turbulence. This is fine, in general, but may prevent natural behavior of particles that can actually reach dry cells, if I well understood the implemented approach. This should be discussed in more detail, as stated in the specific comments.

Please, find below my specific comments and a few typos to be corrected.

**ANS**: The authors are grateful for the thorough and constructive comments. The suggestions have been very helpful and have significantly contributed to improving the quality and clarity of the manuscript. All responses and changes made to the manuscript have been marked in colour for ease of reference. Moreover, Figures 1 and 9 have been revised to follow colourblind-friendly formats.

**Specific comments**

Line 44: "are designed for coastal scenarios where challenging wet-dry transitions do not occur". This sentence appears counterintuitive. Generally, on the coast, waves give rise to wet-dry scenarios. Please, specify what you mean.

**ANS**: The authors appreciate the reviewer's observation. The authors agree that wet-dry transitions are indeed common in coastal scenarios. The sentence has been revised.

System of equations 9: Please specify that the system of equations (9) represents the discretization of the system of equations (6). Additionally, you could explicitly describe the connection between  $v_{\text{disp}}$  in system (6) and the random-walk model in system (9).

**ANS**: The authors are grateful for the suggestion. The sentence has been included to define the system of equations 9. Moreover, the expressions for the velocity components induced by dispersion have been included to describe the connection between  $\mathbf{v}_{\text{disp}}$  in system (6) and the random-walk model in system (9).

Lines 188-190: The time step of the particle is not necessarily an exact divisor of the hydrodynamic time step, as shown also in Figure 2b. In Eq. (15), the particle time step is different from the one computed by Eq. (14)? If so, how is the index "m" imposed? And, please check Eq. (15) versus Eq. (16), because they seem inconsistent (if Eq. (15) is correct, the summation of the particle time step is equal to the hydrodynamic time step, so the hydrodynamic time step minus the summation should be equal to zero in Eq. 16).

ANS: The authors are grateful the observation and apologize for the error. The expression (16) has been modified, and the index "m" has been described in detail in the revised manuscript: Since particles can travel through at most three cells, the number of subdivisions M satisfies  $1 \le M \le 3$ . The value of M is determined individually for each particle based on the flow properties in the cells it traverses: M=1 if the particle remains within the initial cell (Figure 3a), M=2 if it crosses into a neighboring cell (Figure 3b), and M=3 if it travels through three cells (Figure 3c), though this case is uncommon as it requires a specific combination of particle location and velocity field characteristics.

Lines 227-228: This restriction appears quite strict. It appears to limit the effect of turbulence. Are the effect of such limitation discussed? Are the Authors planning to remove it in a later version of the code? Furthermore, is it possible for transported objects that reach a dry cell to stop there?

**ANS**: The authors are grateful for the comment. This restriction is implemented for several reasons:

- It is physically inconsistent for a particle with negligible mass to enter a dry cell. The underlying principle is that particles with significant mass could use inertia to traverse dry regions. Therefore, this restriction specifically applies to massless or negligible-mass particles.
- The turbulence term can displace the particle vertically upward relative to its previous position, resulting in an unphysical "jump" in the particle trajectory. This occurs because dispersion terms depend on friction velocity without properly accounting for flow direction constraints.

These explanations have been included in the revised manuscript.

Figure 4: The logical connection between modules is not fully clear. The figure shows a "Lagrangian particle transport" module (LPT), while in the text it is referred to as "Lagrangian model". Using the same term in the text and in the figure would help the reader, also in Fig. 5. Finally, what do the Authors mean with "Lagrangian model for distributed computations" (lines 249-250)?

**ANS**: The authors are grateful for the observation. The expression "Lagrangian model" has been modified to "LPT model". Moreover, the sentence "Lagrangian model for distributed computations" has been expanded to improve clarity:

The LPT model is currently being implemented to support distributed computations both on multiple CPU-nodes (note that shared memory CPU parallelisation is achieved via OpenMP) and on multi-GPU systems, following the approach used in other SERGHEI modules.

Figure 6 is not clear: are all the three errors normalized by the Euler error? Apparently not (the Euler error should be one), and this is in contrast with both the vertical axis in the figure and the figure caption. The strong dependency on the domain discretization is not clear, either. It appears clear from the MAE and RMSE written in the figures, but not for the graphs. Please, consider to modify this figure.

**ANS**: The authors are grateful for the suggestion and agree that the results were not presented in a sufficiently clear format. Therefore, Figure 6 has been revised to present the errors using a more standard and readable format. Moreover, a reference line with first-order slope has been included to highlight the expected convergence behavior and to facilitate comparison with the observed error decay.

Figure 8: please consider changing the particles color or zooming in the image to make the particles more visible.

**ANS**: The authors are grateful for the suggestion. The particle size has been increased, and a higher zoom level has been applied to the buildings area.

Caption of Figure 10: 10000 particles are reported. This appears in contrast with line 309, where it is written that the simulation was performed with 1000000. Please, check.

**ANS**: The authors apologize for the error. The correct number of particles is  $10^5$ . This number has been modified in the revised version.

**Line 344: Can the Authors clarify what they mean by "areas of stagnant transport"?**

**ANS**: The authors are grateful for the comment. The sentence has been revised to describe more precisely that these regions correspond to areas where particles remain approximately at rest for prolonged periods due to low velocities or topographical constraints. The expression "stagnant transport" has been removed to improve clarity:

Furthermore, this histogram reveals regions where particles tend to accumulate or remain at rest for extended periods, typically due to reduced flow velocities or topographical barriers, which may indicate zones of storage or potential sediment deposition.

**Line 348-349: Can the Authors clarify what they mean by "the higher frequency of particles compared to the event 2"?**

**ANS**: The authors have modified the sentence to improve clarity:

In event 1, the longer travel times observed for most particles in Figure 17a, where particles tend to remain approximately at rest for extended periods, suggest intermittent flow conditions, possibly due to intermittent rainfall, leading to temporary particle deposition.

Conclusions: Nothing is said about model's future developments. Are the Authors planning to include a strategy to account for particles deposition? This also depends on the type of particles that they are aiming to model (plastics, seeds...). The work would possibly benefit from a more critical analysis of the potential applications of the model.

**ANS**: The authors are grateful for the suggestion. The following sentences have been included in the Conclusions section of the revised manuscript:

Future work will focus on optimizing the model to further reduce computational costs and implementing multi-GPU simulations to leverage the capabilities of the SERGHEI hydrodynamic model. Moreover, several enhancements will be incorporated into the LPT module to increase the realism of particle trajectories. These improvements will enable representation of pollutant transport (e.g., microplastics) and biological dispersal (e.g., seeds). Additionally, incorporating particle mass, volume, and inertia will allow modeling of vertical movement, deposition processes, and macroscopic objects such as wooden logs or urban debris in floods. However, accurate simulation of these transport phenomena requires development and validation of specific physical processes that are currently beyond the scope of the present model.

**Typos**

Line 175: I guess there is a typo, as "respectively" is repeated twice.

**ANS**: The authors are grateful for the observation and the word "respectively" has been removed.

Caption of Figure 3: "final position" seems unnecessary.

**ANS**: The authors apologize for the error and the expression "final position" has been removed in the revised version.

Line 266: It should be " $L_1$  norm", and not " $L_1$  Norm". The same for  $L_2$  norm at line 270.

**ANS**: The authors are grateful for the correction. The notation for  $L_1$ -norm and  $L_2$ -norm in the text has been uniformed to be consistent in the manuscript.

Line 283: "described" seems unnecessary.

**ANS**: The suggestion has been included and the word "described" has been removed.

Line 284: "modying" should be "modifying"?

**ANS**: The authors are grateful for the correction and it has been included in the revised manuscript.

Line 300: another way of presenting the L norm is used. Please be consistent in the terminology (choose between L Norm, L norm or L-norm).

**ANS**: The authors are grateful for the observation. The notation for  $L_1$ -norm and  $L_2$ -norm in the text has been uniformed to be consistent in the manuscript.

Line 332: "This" and not "these figure".

**ANS**: The authors are grateful for the correction and it has been included in the revised version.

**Line 339: Please, use the same term, "travelled distance" or "covered distance" for higher consistency.**

**ANS**: The authors are grateful for the suggestions and only the expression "travel distance" is used to have higher consistency.

**Line 347: "the difference between the events is higher"**

**ANS**: The authors are grateful for the correction and this has been included in the revised version.

The authors sincerely thank the reviewer for the careful reading of the manuscript and for pointing out typographical errors. In addition to the corrections suggested by the reviewer, the authors have thoroughly revised the manuscript and corrected other minor typographical issues present in the original version.

**Reviewer 2**

The authors present a Lagrangian model for passive particles coupled to a 2D shallow water model (SERGHEI). In particular, the paper analyzes the accuracy of three different schemes: online 4th order RK, online 1st order Euler and online 4th order RK. For this the authors consider four test cases: a steady vortex, a flow resulting of damn break that collides with some buildings, a channel with cavities, and realistic runoff flow after two precipitation events. For some of the test cases, the particles move only due to advection and for others by both advection and subgrid diffusion (due to unresolved turbulence). The main conclusions are that 1) the model performs well and 2) the Euler scheme gives the best trade-off between accuracy and computational efficiency.

Overall, I find that the work interesting and the model seems to indeed perform well. However, I find the discussion many times superficial, inaccurate, and confusing. I hope that the comments below will help the authors improve their manuscript.

As a disclaimer, I want to mention that I was already preparing this review when I was notified that the other reviewer had upload their comments. I have still finished my review without looking into the other reviewer's comments to avoid bias. However, I have read the comments after finishing to avoid possible repetition or contribute further to the already ongoing discussion. Still, it seems that we have quite different concerns.

**ANS:** The authors thank the reviewer for their thoughtful and detailed comments. In response, the manuscript has been thoroughly revised to improve the clarity of the discussion, the depth of the analysis, and the overall presentation of the conclusions. The authors believe these revisions have significantly strengthened the work.

**Major comments**

1- Lines 37-38. The Lagrangian approach does not, in general, offer detailed insights into processes like deposition, fragmentation, and degradation. This is only the case if such processes are implemented. The main difference between the Lagrangian and the Eulerian approach is that the Lagrangian approach provides insight into the pathways linking the origin to the destination of individual particles.

**ANS:** The authors are grateful for the suggestions, and these have been included in the revised version of the manuscript:

The Lagrangian approach primarily provides information on the pathways linking the origin to the destination of individual particles. It may also capture specific processes affecting debris, such as deposition, fragmentation, and degradation, provided that these processes are explicitly implemented.

2- The discussion about research on Lagrangian transport in coastal environments (l. 43-48) seems inappropriate. I think that there are certainly differences in the numerical approach and maybe the physics of the problem (typical velocities or time scales?) but drying and flooding occurs over vast extents in some coastal systems. The authors can see for example the work by Cucco et al. [1] or recent work by Fajardo-Urbina et al. [2,3] for passive particles transported by depth-averaged flows over regions that flood and dry twice a day! Furthermore, they used offline methods, so lack of flooding and drying is not the reason for using them. I think that one of the main differences is that in these coastal studies the flow of interest changes with a typical time scale that is much longer than the time step needed to advance the particles. Notice that it is common to use temporal interpolation besides using RK4 [4].

**ANS:** The authors appreciate the reviewer's comment. The text have been revised to incorporate it:

In recent decades, numerous computational models have been developed to simulate particle transport. However, the majority of these models are designed for ocean and sometimes coastal scenarios with fixed wet/dry boundaries (Lebreton et al., 2012; Liubartseva et al., 2018). Furthermore, some models update particle positions only at specific time intervals rather than at every time step, in order to reduce the high computational cost (Finaud-Guyot et al., 2023). These so-called offline methods are often used in coastal environments where the flow evolves on time scales much longer than the particle time step, making temporal interpolation feasible (Cucco et al., 2009; Fajardo-Urbina et al., 2023, 2024). It is important to note that flooding and drying occur in many coastal systems, such as estuaries or tidal flats, and offline methods have still been successfully applied in such contexts. The inaccuracy introduced by not updating particle positions at every step is often mitigated by using higher-order schemes, such as a fourth-order Runge-Kutta method (García-Martínez and Flores-Tovar, 1999).

**3- Lines 63-64. The sentence "In this context, two-dimensional models ... " needs to specify the application. This is not the case in general.**

**ANS:** The authors are grateful for the suggestion. The applications have been included in the revised manuscript:

Finally, in order to be effective, the computational model must be both accurate and computationally efficient. This balance is especially important in applications such as flood forecasting, real-time decision support, environmental impact assessments, and large-scale scenario simulations, where timely and reliable results are essential. In this context, depth-averaged two-dimensional hydrodynamic models have proven to offer a favorable trade-off, providing sufficient accuracy for many surface water flow scenarios while incurring significantly lower computational costs compared to fully three-dimensional models (Vacondio et al., 2016; Echeverribar et al., 2019).

4- Lines 100-106. The discussion about the vertical position of the particles is inconsistent. First, the equation of  $z_p$  in (5) is not correct. The particle position has a vertical velocity equal to the velocity of the free surface. In fact, the authors later say that  $z_p = h + z_b$ , so  $\frac{dz_p}{dt} = \frac{dh}{dt}$ . Even then, this is still inconsistent with the rest of the problem, because the particles are carried by a depth averaged flow, which is different than the flow at the free surface. In fact, the depth averaged flow is a mathematical construction so that particles transported by it have no vertical position.

**ANS:** The authors thank the reviewer for this insightful comment. The authors agree that, within the framework of a depth-averaged flow, particles do not possess a physically meaningful vertical velocity component, and the assumption  $\frac{dz_p}{dt} = 0$  is not consistent with the notion of particles following the free surface. To resolve this, the manuscript has been revised to clarify that the vertical position  $z_p$  is not derived from the governing equations but is instead assigned for practical and visualization purposes as:

$$z_p = h(\mathbf{x}_p) + z_b(\mathbf{x}_p)$$

This approach ensures that particles are located at the free water surface, which is especially helpful in domains with variable topography and avoids artifacts caused by unrealistic particle movements in the vertical direction. The authors have also revised the text to clarify that, since the hydrodynamic model is depth-averaged, vertical velocities are neglected, and the particles are advected using the horizontal components of the depth-averaged velocity field:

$$\begin{cases} \frac{dx_p}{dt} = u(\mathbf{x}_p) \\ \frac{dy_p}{dt} = v(\mathbf{x}_p) \\ z_p = h(\mathbf{x}_p) + z_b(\mathbf{x}_p) \end{cases}$$

As observed, the vertical position is unaffected by the flow velocity due to the depth-averaged SWE approximation, which neglects the vertical velocity component. Consequently, the particle is assumed to reside at the free surface, computed as the sum of the water depth h and the bed elevation  $z_b$ . This assignment is not derived from the governing equations but serves primarily for visualization purposes and to maintain numerical robustness. Since the hydrodynamic model is vertically averaged, and thus does not resolve vertical flow structure, particles transported by it do not possess a true vertical coordinate in the physical sense. However, assigning them a position at the free surface ensures consistency with the surface flow and avoids issues arising from irregular bathymetry. Notably, this choice prevents numerical artifacts, such as particles unrealistically crossing obstacles or walls due to inconsistent vertical velocities. Moreover, because the advection velocity is evaluated at the horizontal location of each particle, aligning all particles to the free surface provides a coherent reference for computing motion in the horizontal plane.

**5- Line 119. Turbulence is not a quantity so it cannot be proportional to velocity.**

**ANS:** The authors appreciate the reviewer's observation and apologize the error. The sentence has been removed in the revised manuscript.

6- Figure 1. I find figure 1 very confusing. I really don't understand why/how the particle would follow the green path. It looks also quite different than in Figure 3c. Furthermore, the vector on the cell to the left of the obstacle does not seem right because it would transport particles into the obstacle.

**ANS:** The author thank the reviewer's feedback on Figure 1. The figure has been revised to address the concerns raised:

- The green particle path has been corrected and now more clearly reflects the intended trajectory, consistent with the flow field and the behavior of the algorithm.
- The velocity vectors have been revised to ensure they are physically consistent with the presence of the obstacle. In particular, the vectors to the left, right, above, and below the obstacle have been adjusted so that they no longer incorrectly point toward the obstacle.

7- Section 4.1. The authors do not give sufficient information to reproduce the results. Particularly, the shape of the vortex, the location of release, the velocity of the vortex. The fact that the authors only considered an offline method updated every five hydrodynamic time steps seems restrictive. What if the there is a better trade-off when updating every 3 time steps? In addition, I find figure 6 close to useless. In the caption, it is mentioned that the error is normalized by the Euler error, but it is actually normalized by the RK4 error. By doing this, all the information about how the RK4 error depends on  $\Delta x$  is lost. I would suggest plotting lines in a log-log plot without normalizing. Are the errors scaling as they are supposed to?

**ANS:** The authors appreciate the reviewer's insightful and constructive comments. In response:

- Additional information has been added in Section 4.1 to ensure reproducibility of the test case. The authors now specify that the setup consists of a steady circular vortex with a diameter of 10 meters, centered in the middle of the domain. The vortex velocity field increases radially from zero at the center to a maximum of 100 m/s at the edge (i.e., 10 meters from the center). The particle release location has also been clarified in the revised manuscript.
- Regarding the update frequency of the offline method, the authors acknowledge the potential value of testing additional update intervals. However, preliminary experiments showed that reducing the update frequency to 2 or 3 hydrodynamic time steps resulted in only marginal reductions in computational cost, while offering limited contrast in accuracy compared to the 5-time-step case. Therefore, the 5-step update was chosen to better illustrate the trade-off between computational efficiency and accuracy.

• Figure 6 has been revised following the reviewer's suggestion. Moreover, a reference line with a first-order slope has been added to highlight the expected convergence behavior and to facilitate comparison with the observed error decay.

**8- Section 4.2. The authors say that this is a well-known test case, but they do not test much or compared against any other results.**

ANS: The authors thank the reviewer for the observation. The test case presented in Section 4.2 is a well-known benchmark in hydrodynamics and has been previously simulated and validated using the SERGHEI framework. Specifically, in Caviedes et al. (2023), the computational results obtained with SERGHEI were quantitatively compared against experimental measurements, demonstrating good agreement. The beginning of the Section 4.2 has been revised accordingly. We would like to clarify that the experimental results do not provide a direct validation for transport of particles.

9- Section 4.3. Again, there is no benchmark. I agree that it is a good sign that the results remain symmetric, but this is not a proof that the code is doing everything fine. It is just a proof that there are no asymmetric errors. Furthermore, it is clear in both 4.2 and 4.3 that the diffusive terms are doing something, but it is not shown that what they are doing is correct.

ANS: The case is not intended as formal verification benchmark but rather as a demonstration of the effects of the dispersive random-walk model. Indeed, there is no benchmark. The case is motivated by experimental designs proposed to investigate microplastic entrapment, but no measurements are yet available. The results of this case only intend to show that the dispersive model has an effect on particle distribution and trajectories, and that the distribution symmetry shows a reasonable behaviour. We do not claim that this formally proves correctness, but it provides soft evidence of it in a reasonably complex setting, which is still simple enough to qualitatively assess it. We now clarify this in the text. Furthermore, a comparison with a coarser grid has been included to support the discussion on the importance of high resolution in the Eulerian solver, its impact on Lagrangian particle trajectories, and the interplay between grid resolution and the effects of the dispersive model in the LPT module.

**10- Section 4.4. I find this section interesting as a nice application, but there is some unbalance between the number of figures and the analysis. I find it also strange that for this section the scheme used is not mentioned.**

**ANS:** The authors appreciate the reviewer's positive remark about the application presented in Section 4.4. In response to the concerns raised, the section has been revised by reducing the number of figures and enhancing the depth of the analysis, ensuring a better balance between visual content and discussion. Finally, regarding the numerical scheme, only the Euler method was used, as in the previous test case. The following sentence has been added to the revised manuscript:

As in the previous test case, the Euler method was exclusively used for the simulation of the Arnás case, based on the optimal balance between accuracy and computational efficiency observed in the initial test cases.

11- Finally, the authors do not really justify their conclusion that the Euler scheme gives the best trade-off between accuracy and computational efficiency. A more careful explanation of what they mean and how they reach their conclusion is necessary. At the moment, it remains somewhat subjective in the sense that the error does not seem much larger than for RK4, but it is more efficient, so I can leave with the error.

**ANS:** The authors thank the reviewer for this valuable observation and agree that the justification regarding the choice of the Euler scheme could benefit from further clarification. The revised manuscript now provides a more detailed explanation of how this conclusion was reached.

In Section 4.1 (steady case), the results obtained using the Euler online and RK4 online methods are very similar, with negligible differences in accuracy. In the transient case (Section 4.2), the differences between the two schemes are slightly more noticeable. However, the main limitation of RK4 online lies in its computational cost. For large-scale scenarios involving a high number of particles (on the order of  $10^5$  or more), the RK4 online method can be up to 17 times more computationally expensive than the Euler online method (Table 1 of section 4.2). Given that the purpose of the tool is to support large-scale simulations, the use of RK4 online becomes impractical despite its marginal accuracy advantage. Therefore, the Euler online method was chosen as the best trade-off between accuracy and computational efficiency. While minor differences with RK4 are acknowledged, the significant improvement in computational performance makes the Euler scheme more suitable for the intended applications.

These comments have been included in the revised manuscript.

**Minor comments**

1- Use scientific notation for the number of particles.

**ANS:** The number of particles is now written using scientific notation.

2- Line 81: "The equations flow" -> "The flow equations".

**ANS:** The sentence has been corrected as suggested.

3- The authors use sometimes  $\boldsymbol{u}$  and sometimes  $\boldsymbol{v}$  to denote the velocity. I suggest being consistent.

**ANS:** The velocity is now consistently denoted using v throughout the manuscript.

**4- Line 175. Define A** =  $(A_x, A_y)$  and  $q = (q_x, q_y)$ .

**ANS:** The variables **A** and **q** have been defined as requested.

5- Use italics (math) x and y throughout the paper when referring to coordinates.

**ANS:** The coordinates are now consistently written in italics.

6- Line 334-335. This sentence can join the previous paragraph. Also, specify what is meant with overhead of 2.39 and 1.10. I guess that you mean "Increase ratio" as in Table 1.

**ANS:** The sentence has been merged into the previous paragraph, and the term "overhead" has been replaced with "increase ratio" for consistency with Table 1.

---

## Referee Report (RR1)

**Review (version 2)**

**SERGHEI v2.1: a Lagrangian Model for Passive Particle Transport using a 2D Shallow water Model (SERGHEI-LTP)**

By Vallés et al.

I have read with care the new version of the paper and the point-by-point response to my earlier comments. In general, the authors have satisfactory responses for them and have made the necessary changes in the text. However, I still have a suggestion related to my previous comments 8 and 9. In those comments, I remarked that "test cases" in Sections 4.2 and 4.3 were not really used to analyze the accuracy of the numerical schemes and the algorithm implementation, which is stated as a primary objective. The author's response was that, in particular, for the case in section 4.3, it was not a "formal verification benchmark". Hence, I would suggest that the authors rephrase the sentences in lines 5-8 and 71-76. It would seem to me that the primary objective of this paper is to introduce the capabilities of this new Lagrangian particle model and that, as part of that, you analyze the accuracy and computational efficiency of the numerical schemes and algorithm.

In addition, I have a couple of minor comments:

Line 196. Change "articles" to "particles"

Line 247. It reads "The turbulence term can displace the particle vertically upward relative...". I was confused there because the particles are always on the "surface", so there should be no vertical motion caused by the turbulence. Could you clarify/correct?

---

## Referee Report (RR2)

The authors exhaustively answered previous comments, improving the paper clarity.

Unfortunately, Figure 14 is not visible in the present version, and there are some formatting typo (e.g., line 384).

Please, check and correct such issues.

---

## Author Response (AR2)

Dear Editor,

Please find enclosed the files corresponding to the revised version of the manuscript titled SERGHEI v2.1: a Lagrangian Model for Passive Particle Transport using a 2D Shallow Water Model in SERGHEI (ref: egusphere-2025-722). The manuscript has been modified and improved according to the reviewers' suggestions. The specific ways in which their comments have been addressed are listed below.

Best regards,

The Authors

**Reviewer 1**

The authors exhaustively answered previous comments, improving the paper clarity. Unfortunately, Figure 14 is not visible in the present version, and there are some formatting typo (e.g., line 384). Please, check and correct such issues.

**ANS**: The authors thank the reviewer for the positive comments. We apologize for the formatting issues in the marked revised version. These issues (see images below) were caused by the difftex process, which introduced formatting typos. Importantly, these typos do not appear in the unmarked revised version, where Figure 14 and the rest of the manuscript are correctly displayed.

**Reviewer 2**

I have read with care the new version of the paper and the point-by-point response to my earlier comments. In general, the authors have satisfactory responses for them and have made the necessary changes in the text. However, I still have a suggestion related to my previous comments 8 and 9. In those comments, I remarked that "test cases" in Sections 4.2 and 4.3 were not really used to analyze the accuracy of the numerical schemes and the algorithm implementation, which is stated as a primary objective. The author's response was that, in particular, for the case in section 4.3, it was not a "formal verification benchmark". Hence, I would suggest that the authors rephrase the sentences in lines 5-8 and 71-76. It would seem to me that the primary objective of this paper is to introduce the capabilities of this new Lagrangian particle model and that, as part of that, you analyze the accuracy and computational efficiency of the numerical schemes and algorithm.

**ANS:** The authors thank the reviewer for the positive comments and for the suggestion. Following the recommendation, the text in lines 5-8:

The primary objective of this work is to analyze the accuracy and computational efficiency of the numerical schemes and the algorithm implementation for particle transport.

has been revised to:

The primary objective of this work is to present the capabilities of the new Lagrangian particle model, while also providing an analysis of the accuracy and computational efficiency of the numerical schemes and their implementation for particle transport.

Similarly, the text in lines 71-76:

The primary objective is to analyze the accuracy and computational efficiency of the numerical scheme and the algorithm implementation for particle transport.

has been modified to:

The primary objective is to present the capabilities of the new Lagrangian particle model, together with an accompanying analysis of the accuracy and computational efficiency of the numerical scheme and its implementation for particle transport.

These modifications clarify that the main goal of the paper is to introduce the new Lagrangian particle model, while also providing an analysis of the numerical schemes and algorithm implementation, in line with the reviewer's suggestion.

In addition, I have a couple of minor comments:

Line 196. Change "articles" to "particles".

**ANS:** The authors thank the reviewer for the observation. The typo has been corrected.

Line 247. It reads "The turbulence term can displace the particle vertically upward relative...". I was confused there because the particles are always on the "surface", so there should be no vertical motion caused by the turbulence. Could you clarify/correct?

**ANS:** The authors thank the reviewer for the observation and apologize for the misunderstanding. As noted, the particles are always on the water surface. The paragraph has been rephrased for clarity:

The dispersion term can relocate the particle horizontally across the surface, and if this displacement places it in a dry cell with a higher elevation than its previous location, the particle undergoes an unphysical "jump" in its trajectory. This arises because dispersion terms depend on friction velocity without properly accounting for flow direction constraints.